# Segmentation-Informed Captioning: A Multi-Stage Pipeline for Surgical Vision–Language Dataset Generation

**Mohamed Hamdy**[1]                                                                     MM1905748@QU.EDU.QA

**Fatmaelzahraa Ali Ahmed**[2]                                             FATMAAHMED.HMC@GMAIL.COM

**Mariam Ahmed**[3]                                                                        MA2004988@QU.EDU.QA

**Mohannad AbuHaweeleh** [3]                                                        MA1908120@QU.EDU.QA

**Muraam Abdel-Ghani** [2]                                           MURAAM.ABDELGHANI@OUTLOOK.COM

**Muhammed Arsalan** [4]                                            MUHAMMAD.ARSALAN@QU.EDU.QA

**Abdulaziz Al-Ali** [1,4]                                                                   A.ALALI@QU.EDU.QA

**Shidin Balakrishnan** [2]                                               SBALAKRISHNAN1@HAMAD.QA

[1] *Computer Science and Engineering Department, College of Engineering, Qatar University, Doha, Qatar*

[2] *Department of Surgery, Hamad Medical Corporation, Doha, Qatar*

[3] *College of Medicine, Qatar University, Doha, Qatar*

[4] *KINDI Computing Research Center, College of Engineering, Qatar University, Doha, Qatar*

**Editors:** Accepted for publication at MIDL 2025

## Abstract

Developing models that understand surgical scenes across different procedures and tasks is critical for advancing generalizable surgical AI. Existing approaches often rely on vision-language models (VLMs), but their performance is limited by the quality of available datasets, which are noisy or misaligned–especially those relying on transcribed surgical audio. To address this, we propose a five-stage pipeline to construct more accurate and less noisy vision-language datasets from existing segmentation datasets. Our method applies rule-based heuristics to extract spatial, and interaction cues, which are then used to prompt a large language model (LLM) to produce naturally sound, clinically coherent captions. Evaluation by three medical experts on how well the captions met stage-specific expectations found that 95% of the generated captions scored 3 or higher on a Likert scale.

**Keywords:** Surgical Captioning, Surgical Scene Understanding, Vision-Language Models

## 1. Introduction

Vision Language Models (VLMs) are transforming the surgical domain, especially with the emergence of Large Language Models (LLMs) (Jeong et al., 2024). Although these models have been extensively utilized in Vision Question Answering (VQA) (Ishmam et al., 2024), their capabilities in surgical segmentation and scene comprehension remain mostly unexplored (Seenivasan et al., 2022). To harness the capabilities of LLMs for generating descriptive captions of surgical frames (Jin and Jeong, 2024), a structured approach is essential (Garg et al., 2025). Existing datasets often rely on transcribed surgical audio, which can introduce noise through transcription errors, misalignments, or irrelevant speech–limiting performance on fine-grained tasks like phase or action-triplet recognition (Yuan et al., 2023, 2024b,a). To develop a reliable surgical VLM, high-quality, semantically rich datasets are essential (Li et al., 2024). Motivated by this, we propose a five-stage strategy to generate descriptive captions of the surgical scene employing existing segmentation datasets.

## 2. Methods

**Five-Stage Pipeline** Given an accurate segmentation mask **stage 1** performs object extraction by identifying visible instruments and anatomical structures in each frame. **Stage 2** extends this by associating objects with coarse absolute positions (e.g., "top-left," "center") within the image. **Stage 3** introduces pairwise spatial relationships between objects, such as "to the right of" or "on top of," to capture relative layout. **Stage 4** adds a graded notion of interaction proximity, describing how close instruments are to anatomical targets (e.g., "very close," "touching"), serving as a proxy for action. **Stage 5** shifts to a temporal perspective, aggregating spatial and interaction cues across consecutive frames to produce a clip-level summary describing possible evolving interactions. At each stage, the extracted information is converted into a structured prompt and provided alongside a stage-specific system message to an LLM to generate concise, natural-sounding captions. At each stage, the extracted information from preceding stages–rather than their generated captions–is combined with the current stage's new information and used to construct an LLM prompt. This design ensures modularity and minimizes the risk of error propagation from one stage to the next. The full pipeline is shown in Fig.1, with system prompts detailed in Appendix 4.

**Evaluation Pipeline** To evaluate caption quality, we conducted a blinded assessment where medical experts reviewed captions generated across all five stages by three LLMs: GPT-4o, Deepseek V3, and Llama 3 70B. Fifteen frames were randomly sampled from the EndoVis18 (Allan et al., 2020) dataset for Stages 1 to 4, and for Stage 5, fifteen 10-frame clips were chosen. Each sample was captioned by the LLMs, and outputs were shuffled to prevent bias. Experts rated each caption on a 5-point scale based on the prompt: "How well does the caption meet the overall expectations for this stage?" (See Appendix 4 for specific expectations.)

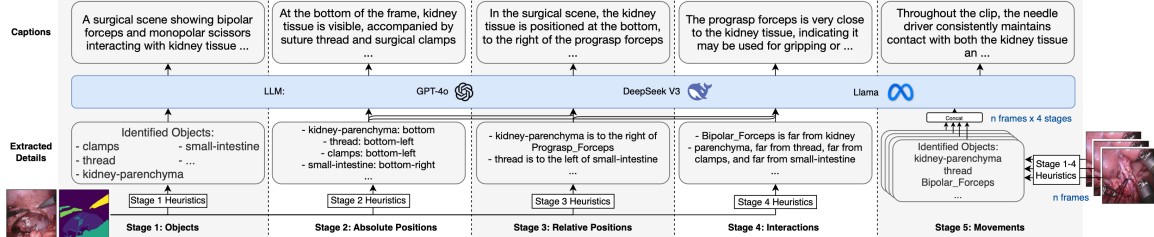

Figure 1: Pipeline for caption generation across the five stages.

## 3. Results and Discussion

**Overall Caption Quality** The distribution of evaluation scores (Fig. 2) shows that most of the captions produced by the five-stage framework received scores between 4 and 5, and low scores (eg. 1 or 2) were rare. More specifically, 95% of all scores were 3 or higher, and 73% were 4 or higher. These results indicate a strong alignment of the proposed five-stage framework with stage-specific expectations and effectiveness in enabling high-quality spatially and temporally grounded training data for fine-grained surgical scene understanding.

**LLM Comparison** Table 1 summarizes the performance of the three LLMs using Friedman ranking and Wilcoxon significance testing. GPT achieved the best average rank (1.97), though with no statistically significant difference ($\alpha = 0.05$). In Stages 1, 4, and 5, Llama

was significantly outperformed by the leading LLMs (GPT in Stages 1 and 4, DeepSeek in Stage 5). No significant differences were found between GPT and DeepSeek, and GPT was never significantly outperformed. These trends are illustrated in the heatmap (Fig. 3), showing GPT consistently near the top while DeepSeek and Llama led in one stage each.

**Stage-wise Caption Quality** A comparison of the stages shows a gradual decline in average caption quality from Stage 1 to Stage 4, as seen in the heatmap (Fig. 3). This decline reflects increasing stage complexity, as earlier stages involve simple object enumeration or localization, while later stages require reasoning about spatial relationships and interactions. Stage 4 received the lowest scores, likely due to its reliance on inferred proximity-based interactions, which can be ambiguous when based on static spatial cues alone. However, performance improves in Stage 5, suggesting that incorporating temporal context across frames helps clarify ambiguity and supports more accurate caption generation.

| | Stage 1 | | Stage 2 | | Stage 3 | | Stage 4 | | Stage 5 | | Overall | |
|---|---|---|---|---|---|---|---|---|---|---|---|---|
| | Rank | p-value | Rank | p-value | Rank | p-value | Rank | p-value | Rank | p-value | Rank | p-value |
| GPT-4o | **1.96** | | 2.00 | 0.564 | 1.98 | 0.914 | **1.84** | | 2.07 | 0.169 | **1.97** | |
| DeepSeek V3 | 2.00 | 0.527 | 2.04 | 0.527 | 2.11 | 0.874 | 1.98 | 0.509 | **1.84** | | 2.00 | 0.867 |
| LLaMA 3.3 70B | $2.04^{\ddagger}$ | 0.042 | **1.96** | | **1.91** | | $2.18^{\ddagger}$ | 0.005 | $2.09^{\ddagger}$ | 0.025 | 2.04 | 0.162 |

**Table 1:** Friedman ranking and Wilcoxon test $p$-values for each LLM across the five captioning stages. All $p$-values are from Wilcoxon tests against the top-ranked model for each stage (shown in bold). The $\ddagger$ symbol marks LLMs significantly worse ($p < 0.05$) than the top performer at that stage.

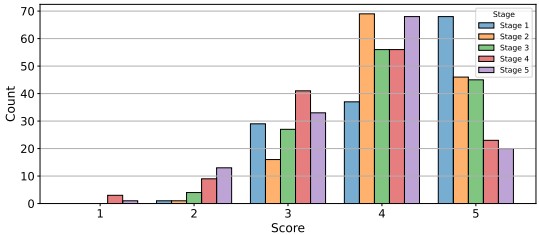

**Figure 2:** Histogram of expert evaluation scores across the five captioning stages.

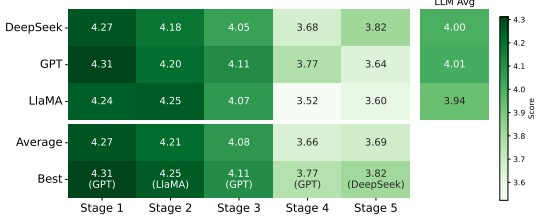

**Figure 3:** Heatmap of average evaluation scores per LLM and stage. The "Best" row highlights the top-performing LLM per stage.

## 4. Conclusion

Our five-stage captioning framework demonstrates strong potential as a source of high-quality, spatially and temporally grounded pseudo-captions to train VLMs on tasks requiring fine-grained understanding of surgical scenes. By incrementally incorporating spatial structure and interaction-level reasoning from segmentation masks, 95% of the evaluated captions scored 3 or higher on a Likert scale. Future work will focus on comparing against captions derived from transcribed surgical audio, evaluating the effect of fine-tuning pre-trained VLMs on our synthetic dataset, and conducting user studies to compare the generated captions with those written by surgeons. We also plan to explore the impact of semantic calibration issues raised in recent work (Wang et al., 2025).

## Acknowledgments

Research reported in this publication was supported by the Qatar Research Development and Innovation Council (QRDI) grant number ARG01-0522-230266. Disclaimer: The content is solely the responsibility of the authors and does not necessarily represent the official views of Qatar Research Development and Innovation Council.

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

- kidney tissue: The internal functional tissue of the kidney, often exposed after the outer layers are removed during surgery, appearing as an irregular soft tissue mass that varies based on exposure and dissection.
- covered kidney: The kidney surface still covered by fascia and perinephric fat; used to distinguish partially obscured anatomical regions, typically seen as a mound-like form covered in fibrous and fatty layers.
- suture thread: Surgical suture material used for stitching tissue; manipulated by instruments but not active itself, appearing as a thin, flexible strand that may curve or stretch taut between points.
- surgical clamps: Surgical instruments used to grip, hold, or compress tissues and vessels during procedures, often appearing as multiple small white objects pressing two or more parts together.
- suturing needle: A needle used in suturing operations, often driven by a needle holder to stitch tissues, typically sharp and curved, resembling a small metallic arc.
- small intestine: Part of the digestive tract, visible in some surgical scenes, typically not interacted with directly in the procedure; it appears as a looped, tubular structure with smooth surfaces.
- bipolar forceps: A robotic instrument that uses bipolar energy to cauterize and manipulate tissue precisely, featuring tapered insulated jaws connected to a shaft and handle.
- prograsp forceps: A da Vinci robotic tool designed for strong gripping, holding, and retracting tissues or organs, with a handle shaft ending in robust, serrated jaws.
- needle driver: A robotic instrument specialized in holding and driving suturing needles through tissue, consisting of a long narrow shaft with a handle at one end and hinged jaws at the other.
- monopolar scissors: Robotic scissors with monopolar cautery capability, used for cutting and dissecting tissue, typically featuring a small curved hook at the tip for precision work.
- ultrasound probe: A drop-in probe used to capture intraoperative ultrasound images for tissue visualization, generally bulky with a flat or convex scanning face.
- suction instrument: A tool used to remove blood, smoke, or fluid from the surgical field to maintain visibility, characterized by a rounded tip with a central opening.
- clip applier: A surgical device that applies metal or plastic clips to blood vessels or ducts for ligation, with tips that include two short, slightly curved jaws.

Figure 4: List of anatomical structures and surgical instruments with a brief description prompted to the LLMs across all stages.

## Appendix A. System Messages

Fig.5 presents the system messages used to prompt each LLM at every stage. These messages were standardized across models to enable consistent comparison. Additionally, each LLM was provided with a list of object classes (anatomical structures and surgical instruments) along with brief descriptions (see Fig.4) to support accurate scene understanding and help the LLM infer more accurate interactions. To evaluate caption quality, expert raters were shown captions generated for each sample and were asked: "How well does the caption meet the overall expectations for this stage?" The stage-specific expectations were:

1. **Stage 1:** Captions simply list what items or instruments are visible in the scene.

2. **Stage 2:** Captions describe where the items are located within the image (e.g., in the middle, near the edge).

3. **Stage 3:** Captions compare items to each other (e.g., one item is to the left or right of another).

4. **Stage 4:** Captions include possible actions or interactions (e.g., an instrument is touching or close to something).

5. **Stage 5:** Captions describe short video clips and should reflect how actions or interactions change over the duration of the clip.

**Stage 1**

You are a nephrectomy surgeon describing a surgical scene from a nephrectomy procedure. Your goal is to describe which instruments and anatomical structures are present in the scene. State what you observe using clear, confident, and factual language. Write a descriptive but concise and structured 1 to 3 sentences caption. Do not add details not explicitly mentioned. Use a simple format like: 'A surgical scene showing [instruments] interacting with [anatomy].'

Possible instruments, devices, and anatomical structures include: {OBJECTS DESCRIPTION}

**Stage 2**

You are a nephrectomy surgeon describing a surgical scene from a nephrectomy procedure. Your goal is to describe where instruments and anatomical structures are located within the frame. Use location phrases like 'in the center', 'on the left side', or 'at the bottom'. State what you observe using clear, confident, and factual language. Write a descriptive but concise and structured 1 to 3 sentences caption. Do not add details not explicitly mentioned. If an object has an empty position set, acknowledge its presence but do not specify its location.Possible instruments, devices, and anatomical structures include: {OBJECTS DESCRIPTION}

**Stage 3**

You are a nephrectomy surgeon describing a surgical scene from a nephrectomy procedure. Your goal is to describe how the instruments and anatomical structures are positioned in the scene, based on both their general locations in the frame and their spatial relationships to each other. Use direction phrases like 'on top of' or 'to the left of' to describe how objects are arranged. State what you observe using clear, confident, and factual language. Write a descriptive but concise and structured 1 to 3 sentences caption. Do not add details not explicitly mentioned.

Possible instruments, devices, and anatomical structures include: {OBJECTS DESCRIPTION}

**Stage 4**

You are a nephrectomy surgeon describing a surgical scene from a nephrectomy procedure. Your goal is to describe how instruments are interacting with anatomical structures or surgical tools, based on how close they appear or whether they are making contact. Use action words such as 'touching', 'grasping', 'approaching', or 'very close to' to describe these interactions. State what you observe using clear, confident, and factual language. Write a descriptive but concise and structured 1 to 3 sentences caption. Do not add details not explicitly mentioned.

Possible instruments, devices, and anatomical structures include: {OBJECTS DESCRIPTION}

**Stage 5**

You are a nephrectomy surgeon describing a surgical scene from a nephrectomy procedure. Your goal is to describe what is happening across a short surgical video clip composed of consecutive frames. Summarize the changes and interactions that occur throughout the duration of the clip. Use action verbs such as 'approaches', 'manipulates', 'remains in contact with', or 'moves toward' to describe how instruments interact with anatomical structures or surgical tools. You may mention spatial positions (e.g., 'on the right', 'center-left') when useful for clarity. State what you observe using clear, confident, and factual language. Write a descriptive but concise and structured caption of 1 to 3 sentences. Do not add details not explicitly mentioned.

Possible instruments, devices, and anatomical structures include: {OBJECTS DESCRIPTION}

Figure 5: System messages used to prompt the LLMs across the five stages. The referenced list of anatomical structures and instruments with their descriptions (e.g., OBJECTS DESCRIPTION) is provided in Fig. 4.

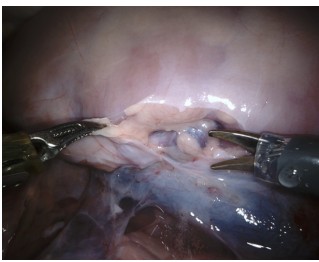

| deepseek-chat | | | | | gpt-4o | | | | | llama3.3-70b | | | | |
|---|---|---|---|---|---|---|---|---|---|---|---|---|---|---|
| | **Caption** | Ev. 1 | Ev. 2 | Ev. 3 | | **Caption** | Ev. 1 | Ev. 2 | Ev. 3 | | **Caption** | Ev. 1 | Ev. 2 | Ev. 3 |
| **Stage 1** | A surgical scene showing bipolar forceps and monopolar scissors interacting with exposed kidney tissue and adjacent covered kidney. | 5 | 4 | 4 | | A surgical scene showing bipolar forceps and monopolar scissors interacting with kidney tissue and a covered kidney. | 5 | 4 | 4 | | A surgical scene showing bipolar forceps and monopolar scissors interacting with exposed kidney tissue and a partially visible covered kidney. | 5 | 4 | 4 |
| **Stage 2** | In the center-right of the frame, exposed kidney tissue is visible, while the covered kidney appears at the top. The bipolar forceps are positioned in the center-left, and the monopolar scissors are located in the center-right. | 5 | 4 | 4 | | In the frame, the kidney tissue is located at the center-right, adjacent to the monopolar scissors, which are positioned for precise dissection. The covered kidney is visible at the top, indicating the area yet to be exposed. On the center-left, the bipolar forceps are poised for tissue manipulation and cauterization. | 5 | 4 | 5 | | In the center-left of the frame, bipolar forceps are visible, while monopolar scissors are located in the center-right, adjacent to exposed kidney tissue. At the top of the frame, a covered kidney is present. The kidney tissue is situated at the center-right. | 5 | 4 | 4 |
| **Stage 3** | The kidney tissue is positioned center-right, below the covered kidney at the top of the frame. The bipolar forceps are held center-left, with the covered kidney situated to their right, while the monopolar scissors are positioned center-right, below the covered kidney. | 5 | 3 | 3 | | In the surgical scene, the covered kidney is positioned at the top, with the kidney tissue located toward its bottom. The bipolar forceps are situated to the left of the covered kidney, while the monopolar scissors are positioned toward the bottom of the covered kidney, near the kidney tissue. | 5 | 2 | 4 | | In the surgical scene, the covered kidney is positioned at the top, with the kidney tissue and monopolar scissors located toward its bottom. The bipolar forceps are situated to the left of the covered kidney, at the center-left. The kidney tissue and monopolar scissors are both centered to the right, with the kidney tissue and monopolar scissors in close proximity to each other. | 5 | 4 | 4 |
| **Stage 4** | The bipolar forceps is grasping the covered kidney near its lower edge, while the monopolar scissors is dissecting the exposed kidney tissue and making contact with the adjacent covered kidney surface. Both instruments are engaged with their respective targets in the surgical field. | 4 | 4 | 3 | | The bipolar forceps are touching the covered kidney, positioned far from the kidney tissue. The monopolar scissors are actively touching both the kidney tissue and the covered kidney, indicating precise dissection and exposure of the kidney. | 4 | 1 | 4 | | The bipolar forceps is touching the covered kidney, which is positioned above the kidney tissue. The monopolar scissors, located near the kidney tissue, is touching both the kidney tissue and the covered kidney, indicating a close interaction with these anatomical structures. The bipolar forceps, however, is far from the kidney tissue, suggesting it is currently focused on the covered kidney. | 4 | 2 | 3 |

Figure 6: Example captions generated by the three employed LLMs for frame 28 of sequence 10 from the EndoVis18 dataset (Allan et al., 2020), shown across the stages. Each row shows the caption produced at a given stage, along with expert scores from three evaluators (Ev. 1–3), based on how well the caption met stage-specific expectations.

## Appendix B. Example Captions and Evaluation

Figures 6–8 show example frames from the EndoVis18 dataset (Allan et al., 2020), along with captions generated by the three LLMs across all five stages and the corresponding scores assigned by each evaluator.

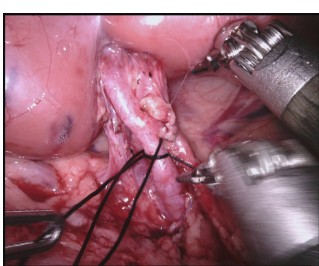

| | deepseek-chat | | | | gpt-4o | | | | llama3.3-70b | | | |
|---|---|---|---|---|---|---|---|---|---|---|---|---|
| | **Caption** | Ev. 1 | Ev. 2 | Ev. 3 | **Caption** | Ev. 1 | Ev. 2 | Ev. 3 | **Caption** | Ev. 1 | Ev. 2 | Ev. 3 |
| **Stage 1** | A surgical scene showing bipolar forceps and monopolar scissors interacting with exposed kidney tissue and adjacent covered kidney. | 4 | 3 | 3 | A surgical scene showing bipolar forceps and monopolar scissors interacting with kidney tissue and a covered kidney. | 4 | 3 | 3 | A surgical scene showing bipolar forceps and monopolar scissors interacting with exposed kidney tissue and a partially visible covered kidney. | 4 | 3 | 3 |
| **Stage 2** | In the center-right of the frame, exposed kidney tissue is visible, while the covered kidney appears at the top. The bipolar forceps are positioned in the center-left, and the monopolar scissors are located in the center-right. | 5 | 4 | 4 | In the frame, the kidney tissue is located at the center-right, adjacent to the monopolar scissors, which are positioned for precise dissection. The covered kidney is visible at the top, indicating the area yet to be exposed. On the center-left, the bipolar forceps are poised for tissue manipulation and cauterization. | 5 | 3 | 5 | In the center-left of the frame, bipolar forceps are visible, while monopolar scissors are in the center-right, adjacent to exposed kidney tissue. At the top of the frame, a covered kidney is present. The kidney tissue is situated at the center-right. | 5 | 4 | 4 |
| **Stage 3** | The kidney tissue is positioned center-right, below the covered kidney at the top of the frame. The bipolar forceps are held center-left, with the covered kidney situated to their right, while the monopolar scissors are positioned center-right, below the covered kidney. | 4 | 3 | 4 | In the surgical scene, the covered kidney is positioned at the top, with the kidney tissue located toward its bottom. The bipolar forceps are situated to the left of the covered kidney, while the monopolar scissors are positioned toward the bottom of the covered kidney, near the kidney tissue. | 4 | 4 | 4 | In the surgical scene, the covered kidney is positioned at the top, with the kidney tissue and monopolar scissors located toward its bottom. The bipolar forceps are situated to the left of the covered kidney, at the center-left. The kidney tissue and monopolar scissors are both centered to the right, with the kidney tissue and monopolar scissors in close proximity to each other. | 4 | 4 | 4 |
| **Stage 4** | The bipolar forceps is grasping the covered kidney near its lower edge, while the monopolar scissors is dissecting the exposed kidney tissue and making contact with the adjacent covered kidney surface. Both instruments are engaged with their respective targets in the surgical field. | 3 | 2 | 4 | The bipolar forceps are touching the covered kidney, positioned far from the kidney tissue. The monopolar scissors are actively touching both the kidney tissue and the covered kidney, indicating precise dissection and exposure of the kidney. | 5 | 3 | 4 | The bipolar forceps is touching the covered kidney, which is positioned above the kidney tissue. The monopolar scissors, located near the kidney tissue, is touching both the kidney tissue and the covered kidney, indicating a close interaction with these anatomical structures. The bipolar forceps, however, is far from the kidney tissue, suggesting it is currently focused on the covered kidney. | 4 | 2 | 4 |

Figure 7: Example captions generated by the three employed LLMs for frame 23 of sequence 2 from the EndoVis18 dataset (Allan et al., 2020), shown across the stages. Each row shows the caption produced at a given stage, along with expert scores from three evaluators (Ev. 1–3), based on how well the caption met stage-specific expectations.

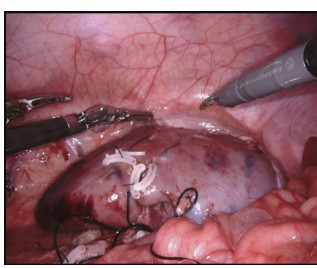

| | deepseek-chat | | | | gpt-4o | | | | llama3.3-70b | | | |
|---|---|---|---|---|---|---|---|---|---|---|---|---|
| | Caption | Ev. 1 | Ev. 2 | Ev. 3 | Caption | Ev. 1 | Ev. 2 | Ev. 3 | Caption | Ev. 1 | Ev. 2 | Ev. 3 |
| **Stage 1** | A surgical scene showing bipolar forceps and monopolar scissors interacting with exposed kidney tissue and adjacent covered kidney. | 5 | 4 | 5 | A surgical scene showing bipolar forceps and monopolar scissors interacting with kidney tissue and a covered kidney. | 5 | 5 | 5 | A surgical scene showing bipolar forceps and monopolar scissors interacting with exposed kidney tissue and a partially visible covered kidney. | 5 | 5 | 4 |
| **Stage 2** | In the center-right of the frame, exposed kidney tissue is visible, while the covered kidney appears at the top. The bipolar forceps are positioned in the center-left, and the monopolar scissors are located in the center-right. | 3 | 4 | 5 | In the frame, the kidney tissue is located at the center-right, adjacent to the monopolar scissors, which are positioned for precise dissection. The covered kidney is visible at the top, indicating the area yet to be exposed. On the center-left, the bipolar forceps are poised for tissue manipulation and cauterization. | 3 | 4 | 5 | In the center-left of the frame, bipolar forceps are visible, while monopolar scissors are located in the center-right, adjacent to exposed kidney tissue. At the top of the frame, a covered kidney is present. The kidney tissue is situated at the center-right. | 3 | 4 | 5 |
| **Stage 3** | The kidney tissue is positioned center-right, below the covered kidney at the top of the frame. The bipolar forceps are held center-left, with the covered kidney situated to their right, while the monopolar scissors are positioned center-right, below the covered kidney. | 5 | 4 | 4 | In the surgical scene, the covered kidney is positioned at the top, with the kidney tissue located toward its bottom. The bipolar forceps are situated to the left of the covered kidney, while the monopolar scissors are positioned toward the bottom of the covered kidney, near the kidney tissue. | 5 | 3 | 4 | In the surgical scene, the covered kidney is positioned at the top, with the kidney tissue and monopolar scissors located toward its bottom. The bipolar forceps are situated to the left of the covered kidney, at the center-left. The kidney tissue and monopolar scissors are both centered to the right, with the kidney tissue and monopolar scissors in close proximity to each other. | 5 | 5 | 4 |
| **Stage 4** | The bipolar forceps is grasping the covered kidney near its lower edge, while the monopolar scissors is dissecting the exposed kidney tissue and making contact with the adjacent covered kidney surface. Both instruments are engaged with their respective targets in the surgical field. | 4 | 1 | 5 | The bipolar forceps are touching the covered kidney, positioned far from the kidney tissue. The monopolar scissors are actively touching both the kidney tissue and the covered kidney, indicating precise dissection and exposure of the kidney. | 4 | 5 | 5 | The bipolar forceps is touching the covered kidney, which is positioned above the kidney tissue. The monopolar scissors, located near the kidney tissue, is touching both the kidney tissue and the covered kidney, indicating a close interaction with these anatomical structures. The bipolar forceps, however, is far from the kidney tissue, suggesting it is currently focused on the covered kidney. | 4 | 5 | 5 |

Figure 8: Example captions generated by the three employed LLMs for frame 26 of sequence 1 from the EndoVis18 dataset (Allan et al., 2020), shown across the stages. Each row shows the caption produced at a given stage, along with expert scores from three evaluators (Ev. 1–3), based on how well the caption met stage-specific expectations.

