# OpenReview forum: "Segmentation-Informed Captioning: A Multi-Stage Pipeline for Surgical Vision–Language Dataset Generation"
_MIDL.io/2025/Short_Papers — MIDL 2025 - Short Papers_

### Official Review · Reviewer_qwVQ · 2025-04-19

**Rating:** 4
**Confidence:** 3

**Summary:**

Given a surgical video segmentation dataset, this paper presents a rule-based framework that uses LLMs to generate a language caption for a surgical clip based on its segmentation masks. The purported aim of this is to better train vision-language models for surgical videos with better captions.

**Strengths:**

- The heuristics used to design the 5-stage auto captioning system seem well-founded and intuitive.
- The experiments perform a user study to assess the utility of the various mainstream LLMs for this purpose.

**Weaknesses:**

- There doesn't seem to be any mechanism for quality control in the autocaptioning system. Given its sequential nature where mistakes in stage i impact stage (i+1), mistakes could compose between the stages, conceivably.
- Recent work ([Wang et al ICLR 2025](https://arxiv.org/abs/2410.04315)) has shown that humans and LLMs alike are miscalibrated when using expressions of certainty. It is conceivable that asking an LLM to distinguish between expressions such as "close" and "very close" as in this work would lead to miscalibrated outputs. This should be addressed and/or discussed in a future version of this paper.

There are also a few missing comparisons that would be crucial for future versions of this work:
- A future version of this paper will need a comparison against training on existing vision-language datasets based on transcribed audio directly, as that is the primary motivator presented in this work. It is currently unclear if the designed heuristics capture everything important in a surgical scene for operational use.
- The actual comparison needed in a user study would be against manually generated captions by a surgeon or a trained expert for that scene.

---

### Decision · Program_Chairs · 2025-05-01

Accept